# Lysosomal Function Impacts the Skeletal Muscle Extracellular Matrix

**DOI:** 10.3390/jdb9040052

**Published:** 2021-11-23

**Authors:** Elizabeth C. Coffey, Mary Astumian, Sarah S. Alrowaished, Claire Schaffer, Clarissa A. Henry

**Affiliations:** 1School of Biology and Ecology, University of Maine, Orono, ME 04469, USA; Elizabeth.Coffey@stjude.org (E.C.C.); salrowaished@kfu.edu.sa (S.S.A.); claire.Schaffer@maine.edu (C.S.); 2Graduate School of Biomedical Science and Engineering, University of Maine, Orono, ME 04469, USA; Mary.Astumian@maine.edu

**Keywords:** zebrafish, spinster, lysosomal myopathy, basement membrane, myotendinous junction, skeletal muscle

## Abstract

Muscle development and homeostasis are critical for normal muscle function. A key aspect of muscle physiology during development, growth, and homeostasis is modulation of protein turnover, the balance between synthesis and degradation of muscle proteins. Protein degradation depends upon lysosomal pH, generated and maintained by proton pumps. Sphingolipid transporter 1 (*spns1*), a highly conserved gene encoding a putative late endosome/lysosome carbohydrate/H^+^ symporter, plays a pivotal role in maintaining optimal lysosomal pH and *spns1^−/−^* mutants undergo premature senescence. However, the impact of dysregulated lysosomal pH on muscle development and homeostasis is not well understood. We found that muscle development proceeds normally in *spns1^−/−^* mutants prior to the onset of muscle degeneration. Dysregulation of the extracellular matrix (ECM) at the myotendinous junction (MTJ) coincided with the onset of muscle degeneration in *spns1^−/−^* mutants. Expression of the ECM proteins laminin 111 and MMP-9 was upregulated. Upregulation of laminin 111 mitigated the severity of muscle degeneration, as inhibition of adhesion to laminin 111 exacerbated muscle degeneration in *spns1^−/−^* mutants. MMP-9 upregulation was induced by tnfsf12 signaling, but abrogation of MMP-9 did not impact muscle degeneration in *spns1^−/−^* mutants. Taken together, these data indicate that dysregulated lysosomal pH impacts expression of ECM proteins at the myotendinous junction.

## 1. Introduction

Proper development and maintenance of skeletal muscle is essential for normal physiology and locomotion. Protein turnover plays a major role in muscle development and homeostasis. For example, artificial selection of chicken lines for fast and slow growth rates suggests that the rate of protein degradation is the major factor underlying overall muscle growth during development in amniotes [1]. Additionally, protein degradation has been shown to underlie muscle growth in most bony fish [2]. Protein degradation mainly takes place in the lysosome, a principal proteolytic system that accounts for more than 98% of long-lived bulk protein turnover in skeletal muscle [3]. The efficiency of lysosomal protein degradation depends upon acidic lysosomal pH. This dependence is because lysosomal protein degradation requires cathepsins that are optimally active at a pH of 4.5–5. Lysosomal pH is generated and maintained by lysosomal proton pumps. Despite the importance of protein degradation in muscle growth during development, the impact of dysregulated lysosomal pH on muscle development, growth, and homeostasis is unknown.

Adhesion of muscle fibers to their surrounding extracellular matrix (ECM) is essential for muscle development and homeostasis [4]. The ECM is not static and undergoes remodeling throughout development and homeostasis, but it is not well understood how lysosomal-mediated protein turnover impacts muscle ECM composition and muscle structure. For example, the isoform of the heterotrimeric protein laminin changes during muscle development in amniotes and zebrafish [5,6]. Laminin 111 is the major constituent during initial muscle development. After initial muscle development laminin 111 is replaced with laminin 211. However, several lines of evidence suggest that there are reciprocal interactions between laminin and the autophagic/endocytic pathways. A laminin receptor, dystroglycan, plays a role in regulating the endocytosis and subsequent targeting to lysosomes for degradation of laminin 111 in mammary epithelial cells [7]. In addition, autophagy is differentially disrupted in two muscular dystrophies that result from mutations in extracellular matrix proteins. Whereas autophagy is increased in laminin α2 mutant mice and inhibition of autophagy rescues the phenotype; autophagy is decreased in collagen VI mutant mice and potentiation of autophagy rescues the phenotype [8,9]. Taken together, the above data clearly indicate that the autophagic/endocytic pathways and cell adhesion interact, but the consequences of altered lysosome function on the extracellular matrix during muscle development and homeostasis are not known. 

Lysosomal pH is generated and maintained by lysosomal proton pumps. Optimal lysosomal pH is maintained by a vacuolar-type H^+^-ATPase (V-ATPase) that pumps protons into the lysosome, and multiple antiporters and symporters that move protons out of the lysosome [10,11,12]. V-ATPase is the primary driver of acidification within the endocytic pathway and the lysosome. Bafilomycin AI (BafA1), a specific inhibitor of V-ATPase, is able to block all lysosomal degradation that depends on the acidic pH, thus illustrating the requirement for V-ATPase in lysosomal protein degradation. Sphingolipid transporter 1 (*SPNS1*), a highly conserved gene that encodes a putative late endosome/lysosome carbohydrate/H^+^ symporter, plays a pivotal role in maintaining optimal lysosomal pH [13]. Disruption of *spns1* has been shown to cause excessive lysosomal accumulation and increased senescence in zebrafish embryos [12]. Here, we used the zebrafish mutant *spns1^−/−^* to investigate the requirement for normal lysosomal pH during muscle development and homeostasis. 

In this study, we showed that dysregulated lysosomal pH disrupts muscle homeostasis and causes muscle degeneration. We found that the ECM protein laminin 111 was abnormally re-expressed at the myotendinous junction (MTF) in *spns1^−/−^* muscle at 3.5 days post-fertilization (dpf). We hypothesize that laminin 111 expression is a compensatory response because double mutants for *spns1* and a laminin receptor, Integrin α7 (Itga7), showed significantly worse muscle degeneration than either single mutant. However, the increased laminin 111 is not sufficient because there is still muscle degeneration in *spns1* mutants. We hypothesized that lysosomal dysfunction decreased muscle membrane stability because this stability would likely not be fully rescued with the increased adhesion to the ECM with upregulated laminin 111. Experiments using Evans Blue Dye to identify membrane damage corroborated this hypothesis. Muscle degeneration in the zebrafish model of Duchenne muscular dystrophy (DMD) also predominantly results from sarcolemmal instability. Given the correlation between DMD-induced muscle damage due to sarcolemmal instability and increased levels of *mmp9* in DMD [14], we investigated *mmp9* expression and detected upregulation of *mmp9* in *spns1^−/−^* larvae. In contrast to the beneficial impact of inhibiting Mmp9 in mouse models of DMD [15], inhibition of *mmp9* expression did not reduce muscle degeneration in *spns1^−/−^* zebrafish. Together, these results suggest that dysregulated lysosomal pH correlates with altered expression of adhesion complex components at the MTJ, and some of these changes partially mitigate the muscle defects.

## 2. Materials and Methods

### 2.1. Zebrafish Husbandry/Mutant/Transgenic Lines

Zebrafish embryos were retrieved from natural spawns of adult zebrafish maintained on 14 h light/10 h dark cycle. *Spns1^hi891/h891^* mutants were a generous gift from the Kishi lab at the Scripps Research Institute. Strains used in this study were *spns1^hi891/h891^* [16], *spns1^hi891/h891^*; *dag1^hu3072^* [17], *spns1^hi891/h891^*; *itga7^−/+^*, and *spns1^hi891/h891^*; *Tg(Nf-**κB:EGFP)^nc1^* [18]. Embryos were grown in embryo rearing media (1× ERM) with methylene blue at 28.5 degrees Celsius and staged according to [19].

### 2.2. Itga7 Mutant Construction

Itga7 crispr mutation was generated following a previously published protocol [20]. The single guide RNA (sgRNA) targeted DNA sequence of the Itga7 gene located in exon2, (NCBI Reference Sequence: XM_017358253.2), was selected by CHOPCHOP web tool (https://chopchop.cbu.uib.no, accessed on 15 April 2015). All sequences of the oligonucleotides used in this study are described in (Table 1). The single guide RNA oligo sequence (sgRNA) with protein were injected into a single cell stage, fertilized, wild type zebrafish embryos resulting in a 5 bp deletion (Figure 1). 

### 2.3. Morpholino Injections

Morpholinos (MOs) were obtained from Gene Tools, LLC. MOs were diluted in sterile water to a stock concentration of 1 mM. Two *itga6* translation-blocking MOs had the following sequences: MO1 5′-AGCTCCATTGCCTGAAATGAATG-3′ and MO2 5′-CTGTTGTATGAAAAATATAGCCCTT-3′. These two MOs were co-injected so that embryos received 9 ng MO1 and 8 ng MO2. 

### 2.4. Cyclopamine (CyA)/Aurintricarboxylic Acid (ATA) Treatment

Embryos were treated with CyA to disrupt Sonic hedgehog (Shh) signaling. CyA dilution was chosen based on [21]. CyA (Toronto Chemical) was dissolved in 100% EtOH to a stock concentration of 5 mM. 250 uL of the 5 mM stock was added to 25 mL of 1× ERM, bringing the final concentration to 50 uM CyA and 1% EtOH. Control embryos were treated with 1% EtOH. 

Embryos were treated with ATA to disrupt TWEAK signaling. ATA dilution was chosen based on [22]. ATA (Sigma–Aldrich, St. Louis, MO, USA) was dissolved in 100% EtOH to a stock concentration of 100 mM. 50 uL of 100 mM stock was added to 50 mL of 1× ERM, bringing the final concentration to 100 uM ATA and 0.1% EtOH. Control embryos were treated with 0.1% EtOH.

### 2.5. Evans Blue Dye (EBD) Injection

Zebrafish were anesthetized in 0.16 mg/mL tricaine in 1× ERM and side mounted on 1% agarose-lined Petri dishes in minimal volume of liquid. EBD (1% *w*/*v*) was dissolved in a 0.9% saline solution. Injection needles were pulled from glass capillary tubes containing a filament on a Sutter Flaming/Brown Micropipette Puller. EBD solution was loaded into injection needles and injected into the circulation of anaesthetized ~72 hpf zebrafish using a MMPI-3 pressure injector from ASI. EBD injected zebrafish were maintained in 1× ERM for approximately 12 h before live imaging.

### 2.6. Phalloidin Staining and Immunohistochemistry

All antibodies (Abs) were diluted in block (5% *w*/*v* bovine serum albumin (BSA) in phosphate buffered saline (PBS) with 0.1% Tween20). Alexa 546 phalloidin (molecular probes) staining involved fixing embryos and larvae in 4% paraformaldehyde (PFA) for 4 h at room temperature (RT), washing 5 times for 5 min each (5 × 5′) in 0.1% PBS-Tween, permeabilizing for 1.5 h in 2% PBS-Triton, washing 5 × 5 and then incubating in phalloidin (1:20) for 1–4 h at RT. Ab staining followed phalloidin staining. Ab staining involved blocking for 1 h at RT, incubating in 1° Ab overnight at 4 °C, blocking for 8 h at RT, incubating in 2° Ab overnight at 4 °C, then washing for 1 h. 1° Abs: anti-Laminin-111 1:50 (Sigma); anti-pY-397-FAK 1:50 (Biosource); anti-beta-dystroglycan 1:50 (Novocastra); anti-MMP-9 (GeneTEX). 2° Abs: GAM/GAR 488, 546, 633 1:200 (Invitrogen).

### 2.7. Comparative qRT-PCR

RNA was isolated from whole embryos at 3.5 dpf via Quiagen’s RNeasy Mini Kit (Germantown, MD, USA). One-step comparative qRT-PCR was conducted with Quanta kit reagents and SYBR^®^ Green. An Mx400 machine was used and the annealing temperature was at 60 °C. Approximately 50 ng of RNA template was used per reaction. We used beta-actin as our normalizing transcript. See Table 1 for a complete list of primer information. See Table 2 for qRT-PCR results.

### 2.8. Imaging

Images were taken on an Olympus Fluoview IX-81 inverted microscope with FV1000 confocal system or a Leica TCS SP8 confocal system using the LASX Software. Embryos and larvae were side-mounted in 80% glycerol/20% PBS and viewed with the 20× or 25× objective. Exposure times were kept constant throughout the imaging of an experiment allowing for the comparison of fluorescent levels between images. Live imaging for EBD experiments involved side mounting ~3.5 dpf zebrafish larvae in 0.4% low melt agarose containing 0.16 mg/mL tricaine and imaging via confocal microscopy.

### 2.9. Birefringence

Birefringence was used to quantify the anisotropy of muscle structure [23]. Zebrafish were anesthetized with tricaine and transferred to glass bottom dishes prior to imaging. A Leica MZ10F stereomicroscope, attached DMC5400 camera, and polarizing filters were used to capture images. Imaging parameters, such as magnification, exposure, and gain were kept consistent throughout the experiment. Mean gray values were calculated from birefringent images in FIJI software by outlining fish three independent times and taking the mean of the gray values. The average mean gray value of each embryo was normalized by dividing by the average mean gray value of the ethanol treated sibling controls.

### 2.10. Analysis and Statistics

NF-*κ*B activity was quantified using mean grey values of fluorescent intensity based on [24]. Fluorescent images were captured at 25× magnification. Fluorescence signal intensity was determined using mean grey values with images converted to a pre-set grey scale (ImageJ). All image capturing and threshold parameters were kept constant between experiments. 

Categorical data were analyzed using R version 3.5.1 using Fisher’s exact test. Immunohistochemistry data was visually scored. Relative fluorescence intensity was assigned a score of 0 (no staining), 1 (weak staining), or 2 (strong staining). 

Quantitative data were analyzed using Prism7. Quantitative statistical comparisons of two groups were conducted using student *t*-tests. Quantitative statistical comparisons of multiple groups were conducted using ANOVA followed by Tukey’s honest significant difference multiple comparisons test. 

## 3. Results

### 3.1. Initial Muscle Development Proceeded Normally But Degeneration Began at 3.5 Days Post-Fertilization

As has been previously shown, *spns1*^−/−^ mutant larvae show increased lysosomes by 3.5 dpf (Figure 1A,B) [12,25,26]. Although beta-galactosidase [16] has been observed in *spns1^−/−^* muscle, the effects of lacking *spns1* on muscle structure have not previously been reported. In addition, it is not known whether muscle development proceeds normally in *spns1*^−/−^ embryos prior to beta-galactosidase expression. We found that initial muscle development proceeded normally, where myotomes in *spns1*^−/−^ mutants were similar to wild-type embryos at 1 dpf (Figure 1C,D). By 3.5 dpf, *spns1*^−/−^ mutants began to display slightly unorganized fibers and myotendinous junctions (MTJs) (Figure 1E,F). Overt degeneration of muscle fibers also began at 3.5 dpf (Figure 1H). Fiber-type (slow-twitch or fast-twitch) susceptibility to muscle damage varies between different muscle-related disorders [27]. Interestingly, both fast-twitch and slow-twitch fibers were damaged in *spns1^−/−^* larvae. There were no significant differences in the proportion of *spns1^−/−^* embryos that displayed fast-twitch fiber detachments vs. slow-twitch fiber detachments (*p*-value = 0.8018) (Figure 1I). The severity of muscle degeneration was also similar: there were no significant differences in the proportion of segments per embryo that displayed fast-twitch fiber detachments vs. slow-twitch fiber detachments (*p*-value = 0.5748) (Figure 1J). This data indicated that both fast-twitch and slow-twitch muscles were damaged in *spns1^−/−^* larvae. However, muscle damage in individual *spns1^−/−^* larvae was usually restricted to one fiber type. The proportion of *spns1^−/−^* larvae that contained only fast-twitch detachments or only slow-twitch detachments was significantly greater than *spns1^−/−^* larvae that contained both fast-twitch and slow-twitch muscle fiber detachments (*n* = 27 *spns1^−/−^* larvae, *p*-value = 0.0023, *p*-value = 0.0003, respectively) (Figure 1I). Thus, although both fiber types were susceptible to damage, fiber detachments in individual *spns1^−/−^* larvae had a tendency to be restricted to a single fiber type. The mechanism mediating this effect is unknown. Taken together, these data suggest that dysregulated lysosomal pH did not impact initial muscle development, but disrupted muscle and MTJ homeostasis. 

### 3.2. Aberrant Upregulation of the Extracellular Matrix Protein Laminin 111 at the MTJ Coincided with the Onset of Muscle Degeneration 

The muscle extracellular matrix is critical for muscle development and homeostasis [28]. Although initial muscle development proceeded normally in *spns1^−/−^* embryos, we observed disrupted muscle and MTJ structure at 3.5 dpf. We next asked when abnormal muscle/MTJ structure first appears in *spns1^−/−^* mutants. In wild-type embryos, skeletal muscle fibers grow as development proceeds (note thicker fibers in panel C1 relative to panel B1). This initial growth also occurred in *spns1^−/−^* mutants (compare G1 to F1). At 3 dpf, skeletal muscle fibers in *spns1^−/−^* mutants appeared quite similar to wild-type fibers. However, just several hours later fibers in *spns1^−/−^* embryos were disorganized and began to degenerate (Figure 1(H1,H2), white arrowhead).

The MTJ also developed normally but was disrupted at 3.5 dpf. The basement membrane protein laminin 111 was highly concentrated at the MTJ during the first couple of days of development in both wild-type and *spns1^−/−^* mutants (Figure 2A,B,E,F). In wild-type embryos, the developmental laminin 111 isoform was then replaced by laminin 211 such that by 3.5 dpf laminin 111 is absent from wild-type MTJs (Figure 2C). This pattern was also observed in *spns1^−/−^* mutants (Figure 2D). However, by 3.5 dpf robust expression of laminin 111 reappeared at the MTJ in *spns1^−/−^* mutants (Figure 2F). We quantified laminin 111 staining by assigning a qualitative score of laminin 111 staining to blinded images. Relative laminin 111 staining at the MTJ in WT controls was significantly weaker at 3.5 dpf compared to 1 and 2 dpf (*p*-values < 0.0001) (Figure 2(C2) vs. Figure 2(A2,B2)). In *spns1^−/−^* mutants the relative laminin 111 staining at the MTJ was significantly weaker at 3 dpf compared to 1 dpf (*p*-value = 0.006) (Figure 2(G2) vs. Figure 2(E2)). In fact, there was no significant difference in the relative laminin 111 staining at the MTJ in *spns1^−/−^* mutants at 3.5 dpf vs. 1 dpf (*p*-value > 0.9999) (Figure 2J). These data indicate that laminin 111 was re-expressed in *spns1^−/−^* muscle at a time when it is not normally expressed in WT controls.

### 3.3. A Different Mechanism Underlies Laminin 111 Re-Expression Than Underlies Laminin 111 Expression during Initial MTJ Development 

Shh signaling is required for normal laminin 111 expression at the developing MTJ [29,30]. As laminin 111 was re-expressed after downregulation in *spns1^−/−^* muscle, we hypothesized that Shh signaling would also be required for laminin 111 re-expression in *spns1^−/−^* larvae. Cyclopamine blocks Shh signaling [31]. Cyclopamine was administered beginning at 2 dpf to avoid disruption of initial muscle development [32]. WT and *spns1^−/−^* embryos were treated with 50 µM cyclopamine (with 1% EtoH) beginning at 2 dpf. Controls were treated with 1% EtOH. In wild-type control embryos treated with EtOH, laminin 111 was not expressed in the MTJ at 3.5 dpf (Figure 3A). Surprisingly, laminin 111 was strongly concentrated at the MTJ in cyclopamine-treated wild-type embryos (Figure 3B). The relative laminin 111 concentrated at the MTJ in WT larvae treated with cyclopamine was significantly stronger than the laminin 111 staining in the WT controls (*p*-value < 0.0001). In contrast, cyclopamine treatment had no effect on laminin 111 at the MTJ in *spns1^−/−^* larvae: there was no significant difference in the relative laminin 111 staining at the MTJ in *spns1^−/−^* larvae treated with cyclopamine vs. control *spns1^−/−^* larvae (*p*-value = 0.1838) (Figure 3C,D). These data indicate that, in contrast to initial MTJ development, Shh was not required for laminin 111 re-expression in *spns1^−/−^* larvae. 

### 3.4. The Laminin Receptor Dystroglycan Did Not Genetically Interact with Spinster

The deposition and polymerization of laminin in the basement membrane is regulated by transmembrane cell-surface receptors [33], raising the possibility that these transmembrane receptors could be required for ectopic laminin 111 expression at 3.5 dpf in *spns1^−/−^* mutants. In addition, mutations in laminin 211 or either of the two main transmembrane receptors that bind to laminin 211, *dystroglycan (dag1)* and *integrin alpha 7 (itga7),* result in muscular dystrophy [34,35]. Although laminin 111 is the developmental isoform, exogenous laminin 111 can decrease muscle degeneration when adhesion to laminin 211 is disrupted [35,36]. We hypothesized that: (1) re-expression of laminin 111 could be a compensatory mechanism to combat muscle degeneration that occurs in *spns1^−/−^* mutants, and (2) either *dag1* or *itga7* could be required for ectopic laminin 111 expression in *spns1^−/−^* mutants. We first tested these hypotheses by generating *dag1; spns1* double mutants. Expression of laminin 111 in *dag1* mutants was similar to wild-type embryos: laminin 111 was robustly concentrated at the MTJ during initial muscle development (Figure 4A) and downregulated by 3 dpf (Figure 4B). This downregulation was maintained at 3.5 dpf (Figure 4C). In contrast, although laminin 111 expression was absent at 3 dpf in *dag1; spns1^−/−^* double mutants (Figure 4E), laminin 111 was re-expressed by 3.5 dpf (Figure 4(F1)). This result indicated that dystroglycan was not necessary for re-expression of laminin 111 in *spns1^−/−^* mutants. In addition, the *spns1^−/−^* phenotype was not exacerbated by lack of dystroglycan. Together, these data suggest that dystroglycan did not genetically interact with Spinster.

### 3.5. The Laminin Receptor Integrin α7 Contributed to Muscle Homeostasis in spns1^−/−^ Mutants

We next asked whether *itga7* was required for muscle homeostasis and laminin 111 re-expression in *spns1^−/−^* larvae by generating *itga7; spns1* double mutants. Laminin 111 was not expressed in *itga7* mutants at 3.5 dpf (Figure 5(B1)). In *itga7; spns1* double mutants, laminin 111 was concentrated at the MTJ at 1 and 2 dpf and downregulated by 3 dpf (not shown). However, by 3.5 dpf laminin 111 was re-expressed and concentrated at the MTJ in *itga7; spns1* double mutants (Figure 5(D1), white arrow). Thus, *itga7* was not required for laminin 111 re-expression in *spns1* mutants. Mutations in Itga7 result in Itga7-linked congenital muscular dystrophy [34]. Muscle degeneration occurs in zebrafish deficient for itga7 [37], and in our *itga7* mutant (Figure 5B, red arrowheads). We next tested whether lack of *itga7* exacerbated muscle degeneration in *spns1^−/−^* mutants. Muscle degeneration was more severe in *itga7; spns1* double mutant larvae (Figure 5D, yellow star). Significantly more segments per embryo exhibited degeneration in *itga7; spns1* double mutant larvae compared to either single mutant (Figure 5E). Degenerated muscle fibers in *itga7; spns1* double mutants are surrounded by laminin 111. This result suggests that *itga7* was necessary for laminin 111 integrity at the MTJ in *spns1* mutants. Together, these data suggest that Itgα7 contributed to muscle homeostasis in *spns1* larvae.

### 3.6. Sarcolemmal Instability in spns1 Muscle Fibers

The above data suggest the hypothesis that adhesion to laminin 111 was upregulated in a compensatory manner in *spns1* mutants. Muscle damage in zebrafish models studied thus far can result from two distinct cellular etiologies: failure to maintain sarcolemmal integrity or failure to maintain adhesion at the MTJ external to the sarcolemma [6]. Muscle fibers degenerate in *spns1* mutants, suggesting that adhesion to laminin 111 is not sufficient to promote muscle homeostasis in *spns1* mutants. We thus hypothesized that muscle degeneration in *spns1^−/−^* larvae was due to sarcolemmal instability. We used cell impermeable Evans blue dye (EBD), which only fills muscle fibers when sarcolemmal integrity is compromised, and beta-dystroglycan (DG) antibody staining to distinguish between these two etiologies. EBD was observed in long, attached fibers in *spns1^−/−^* mutants at 3.5 dpf (Figure 6B,C). This result indicated that sarcolemmal damage occurred prior to overt muscle degeneration. Detection of the membrane-integral protein beta-DG at the detached ends of retracted fibers suggested adhesion failure at the MTJ, while beta-DG retained at the MTJ indicated failure to maintain sarcolemmal integrity [38]. The proportion of *spns1^−/−^* mutants with fiber detachments showing the retention of beta-DG at the MTJ (Figure 6F) was significantly greater than the proportion of *spns1^−/−^* mutants showing the retention of beta-DG at the detached ends of retracted fibers (*p*-value = 0.0002) (Figure 6G). Additionally, the proportion of segments per *spns1^−/−^* mutant with fiber detachments showing retention of beta-DG at the MTJ was significantly greater than the proportion of segments showing the retention of beta-DG at the detached ends of retracted fibers (*p*-value < 0.0001) (Figure 6D). These data indicate that fiber detachments in *spns1^−/−^* mutants predominantly resulted from sarcolemmal instability.

### 3.7. MMP-9 Was Upregulated in spns1^−/−^ Mutants

The above data indicate that while Itga7-mediated adhesion to laminin 111 contributed to muscle homeostasis in *spns1^−/−^* mutants, it was not sufficient because the instability of muscle membranes in *spns1^−/−^* mutants also contributed to muscle degeneration. In the zebrafish model of Duchenne muscular dystrophy (DMD), loss of sarcolemmal integrity precedes the detachment of muscle fibers from the basement membrane at the MTJ [6]. Increased levels of MMP-9 are associated with the pathogenesis of DMD in the *mdx* mouse model, the canine X-linked muscular dystrophy in Japan (CXMD_J_) model, and muscle biopsies from human DMD patients [14,39,40]. The laminin α1 chain of the laminin 111 protein promotes the production and activity of MMP-9 [41,42]. As increased MMP-9 levels are associated with the pathogenesis of DMD, and laminin 111 can stimulate the expression of MMP-9, we asked whether MMP-9 expression was increased in *spns1^−/−^* mutants. MMP-9 was expressed in wild-type embryos and *spns1^−/−^* mutants at 24 hpf (Figure 1(C1–D2)), but was not expressed in wild-type larvae at 3.5 dpf (Figure 7A–A3). Transcription of *mmp9* was upregulated 15.1 fold in *spns1^−/−^* mutants compared to WT controls at 3.5 dpf (Figure 7C, Table 3). This result was corroborated at the protein level: MMP-9 was readily detected in *spns1^−/−^* mutants at 3.5 dpf but not in wild-type larvae (Figure 7B–B3 vs. Figure 7A–A3). Relative MMP-9 staining was significantly stronger in *spns1^−/−^* mutants compared with WT controls at 3.5 dpf (*p*-value < 0.01) (Figure 7D). These data indicate that MMP-9 was increased at the transcription and protein levels in *spns1^−/−^* muscle at a time when MMP-9 was normally not expressed.

### 3.8. Tnfsd12-Fn14 Signaling Axis Regulated Increased MMP-9 Expression

Proinflammatory cytokines are known mediators of skeletal muscle degeneration. The muscle-wasting cytokine TNF superfamily member 12 (tnfsf12) induces MMP-9 expression in skeletal muscle. In MMP-9 knockout mice, tnfsf12-induced muscle degeneration in inhibited, indicating that MMP-9 is required for tnfsf12-induced muscle degeneration (Li et al., 2009). We performed qRT-PCR to determine whether *tnfsf12* expression was increased in *spns1^−/−^* mutants at 3.5 dpf. We found that transcription of *tnfsf12* was upregulated 34.2 fold in *spns1^−/−^* mutants compared to WT controls at 3.5 dpf (data not shown). Next we asked whether Tnfsf12 induces *mmp9* gene expression in *spns1^−/−^* mutants. Tnfsf12 signaling involves the binding of Tnfsf12 to its receptor, fibroblast growth factor-inducible 14 (Fn14) [43]. The small molecule inhibitor aurintricarboxylic acid (ATA) blocks Tnfsf12-induced signaling downstream from the Tnfsf12-Fn14 complex [43]. Although Tnfsf12-Fn14 signaling is not required for normal development in mice, we administered ATA beginning at 2 dpf to avoid any potential disruption of muscle development (Maecker et al., 2005). We treated embryos continuously with 100 µM ATA (with 0.1% EtOH) beginning at 2 dpf and analyzed MMP-9 expression at 3.5 dpf. This dose was based on a previous zebrafish study showing that 100 µM ATA caused no significant damage to embryos, while concentrations above 100 µM ATA caused embryo death [22]. Controls were treated with 0.1% EtOH. MMP-9 was not detected in WT controls treated with 0.1% EtOH or 100 µM ATA (Figure 8A,B). MMP-9 staining was detected in *spns1^−/−^* mutants treated with 0.1% EtOH; however, MMP-9 staining was not readily visible in ATA treated *spns1^−/−^* mutants (Figure 8C,D). 

Tnfsf12 induces gene expression of *mmp9* via the activation of NF-*κ*B and AP-1 transcription factors in skeletal muscle [44]. Thus, we used the *Tg(NF-κB:EGFP)* zebrafish line to ask whether Tnfsf12-induced activation of NF-*κ*B is detected in *spns1^−/−^* mutants [18]. The *Tg(NF-κB:EGFP)* line was crossed into the *spns1^+/−^* line to detect NF-*κ*B activity in *spns1^−/−^* mutants. Previous work from our lab showed that zebrafish continuously exposed to 2% EtOH from 30 hpf–3 dpf exhibit increased NF-*κ*B activity as evidenced by increased GFP expression in muscle fibers [45]. Hence, we used WT embryos exposed to 2% EtOH from 30 hpf–3.5 dpf as a positive control. *Tg(NF-κB:EGFP)*; *spns1^−/−^* mutants (Figure 8(G,G1)), *Tg(NF-κB:EGFP);* WT negative controls (Figure 8(E,E1)), and *Tg(NF-κB:EGFP)*; WT positive controls (Figure 8(F,F1)) were fixed at 3.5 dpf. Muscle fibers in *Tg(NF-κB:EGFP)*; WT negative controls expressed significantly less GFP at 3.5 dpf compared with muscle fibers in *Tg(NF-κB:EGFP)*; WT positive controls (*p*-value < 0.0001) (Figure 8(E1,F1)). Muscle fibers in *Tg(NF-κB:EGFP)*; *spns1^−/−^* mutants also expressed significantly less GFP at 3.5 dpf compared with muscle fibers in *Tg(NF-κB:EGFP)*; WT positive controls (*p*-value < 0.0001) (Figure 8(G1,F1)). There was no significant difference of GFP expression in muscle fibers between the *Tg(NF-κB:EGFP)*; *spns1^−/−^* mutants and *Tg(NF-κB:EGFP)*; WT negative controls (*p*-value = 0.5059) (Figure 8(G1,E1)). These data suggest that Tnfsf12 induced gene expression of *mmp9* independent of NF-*κ*B activation in *spns1^−/−^* mutants. 

### 3.9. Reduced MMP-9 Expression Did Not Impact Muscle Degeneration 

Increased MMP-9 levels are associated with pathogenesis in DMD [14,39,40]. The above data indicate that ATA treatment dramatically reduces MMP-9 levels in *spns1^−/−^* mutants. We thus asked whether reducing MMP-9 levels would improve muscle structure in *spns1^−/−^* mutants. We assessed muscle structure by quantifying birefringence in *spns1^−/−^* mutants. Healthy muscle is an organized structure consisting of attached, parallel muscle fibers which act as a light polarizer. By passing the light through a polarizing filter, a comparison can be made between organized muscle, appearing bright white and damaged and/or disorganized muscle, which appears dark. This difference in order, as measured by birefringence, can be quantified by measuring average mean gray values. We found that there was no significant difference in birefringence (measured as mean gray value) (Figure 8H) in control versus ATA treatment. There was no obvious difference between the number of zebrafish that had normal, mild, or severe muscle damage (Figure 8I) in ATA versus ethanol treated embryos as measured by phalloidin staining of the muscle fibers. 

## 4. Discussion

Modulation of protein turnover is important for muscle development and homeostasis. Protein turnover requires a balance between the rate of protein synthesis and the rate of protein degradation. The efficiency of protein degradation is dependent upon regulation of lysosomal pH. Our data support the hypothesis that dysregulation of lysosomal pH contributes to muscle degeneration. It is known that loss of the carbohydrate/H^+^ symporter spinster (spns1) results in premature senescence [12,16,25], but it was not known whether muscle develops normally prior to muscle degeneration. Here, we show that loss of *spns1* does not affect skeletal muscle development. Muscle degeneration begins approximately 3.5 dpf. At this time, there was misregulation of the extracellular matrix at the myotendinous junction (MTJ). Expression of the developmental isoform of laminin, laminin 111, was observed at MTJs in *spns1^−/−^* mutant embryos. This reexpression of laminin 111 appeared to be a compensatory response, as disruption of the laminin receptor Itga7 exacerbated the phenotype in *spns1^−/−^* mutant embryos. However, re-expression of laminin 111 was not sufficient for muscle homeostasis in *spns1^−/−^* mutant embryos. We hypothesize that this is mainly due to the fact that the main mechanism underlying muscle degeneration in *spns1^−/−^* mutants is membrane instability, which is not improved with increased cell–matrix adhesion. Similar to other degenerative contexts where membrane stability is compromised, MMP-9 expression was upregulated in *spns1^−/−^* mutant embryos. In contrast to these other contexts, inhibition of MMP-9 expression did not reduce muscle degeneration (summarized in Table 4). Together, these data indicate that dysregulated lysosomal pH has pleiotropic effects on zebrafish skeletal muscle, resulting in muscle degeneration.

Although it has been shown that increased SA-beta-galactosidase activity, a marker of senescence, is detected in *spns1^−/−^* zebrafish at 3.5 dpf [16], it was not known if *spns1* is required for initial muscle development. We found that initial muscle development proceeds normally in *spns1^−/−^* mutants prior to overt muscle degeneration at 3.5 dpf. This result suggests that dysregulated lysosomal pH causes muscle damage following primary muscle development in zebrafish and contributes to the growing body of evidence indicating that proper lysosomal function is critical for muscle homeostasis. Dysregulation of lysosomal pH has been shown to cause muscle atrophy induced by the antimalarial drug Chloroquine [46]. Loss of cathepsin D-mediated lysosomal protein degradation, which requires a pH of 4.5–5, has also been linked to muscle atrophy in both human patients and zebrafish [47,48]. Although there is no known human disease linked to spns1, Cathepsin D acts downstream of Spns1: loss of spns1 contributes to impaired cathepsin D activity and increased spns1 promotes cathepsin D activity [13,49]. Interestingly, a previous study showed that EtOH treatment increases lysosomal pH in rat hepatocytes, thereby causing significantly reduced protein degradation [50]. We recently found that ethanol exposure causes muscle damage in zebrafish [45]. It will be interesting in the future to determine if dysregulated lysosomal pH contributes to EtOH-induced muscle degeneration in zebrafish. 

Interactions between muscle cells and their surrounding ECM regulate cellular adaptive responses in homeostasis and disease. Laminin 111 is a principal ECM component that is required for the formation of normal muscle and MTJs during embryonic development [30,51]. After initial muscle development, laminin 111 is replaced with laminin 211 [52]. MDC1A is a muscular dystrophy caused by mutations in lama2 but laminin-111 protein therapy significantly improves pathology of mouse models for MDC1A [53]. We hypothesized that ectopic laminin 111 expression in *spns1^−/−^* larvae at 3.5 dpf was an innate adaptive response to muscle degeneration. One direct approach to test this hypothesis would be to generate laminin apha1, beta1, or gamma1 *spns1^−/−^* double mutants. However, since laminin 111 is required for normal muscle development [30,51], we tested our hypothesis with Integrin α7 mutants because Itgα7 is not required for initial muscle development. After using CRISPR-Cas9 to make Integrin α7 mutants, we then generated *itga7; spns1^−/−^* double mutants and observed increased muscle degeneration compared to single *spns1^−/−^* mutant embryos. 

Matrix metalloproteinases (MMPs), a group of zinc-dependent endopeptidases, are known to cleave all ECM proteins. MMP gene expression and activity is tightly regulated during development and homeostasis; however, changes in MMP expression and activity occur in many diseases [54]. With respect to muscle diseases, it has been shown that MMP-9 plays an important role in the pathogenesis of DMD [39,55]. Given that both *spns1^−/−^* larvae and the zebrafish model of DMD show muscle damage that results from sarcolemmal instability, we asked whether MMP-9 might participate in muscle degeneration induced by loss of spns1. MMP-9 mRNA and protein levels were increased in *spns1^−/−^* larvae compared to WT controls. The cytokine Tnfsf12 has been shown to promote MMP-9 expression and muscle damage in multiple forms of muscle atrophy [56,57,58]. Our data suggest that Tnfsf12 expression mediates increased MMP-9 expression in *spns1^−/−^* larvae because treatment with ATA dramatically decreases MMP-9 expression in *spns1^−/−^* larvae. However, treatment with ATA was not sufficient to reduce muscle damage in *spns1^−/−^* larvae. This result suggests that in contrast to DMD, increased MMP-9 is not a major mechanism underlying muscle degeneration in *spns1^−/−^* larvae. It will be interesting in the future to determine the mechanisms underlying loss of sarcolemmal integrity in zebrafish in response to dysregulated lysosomal pH.

We have addressed the impact of dysregulated lysosomal pH on muscle development in zebrafish. Our data showed that *spns1^−/−^* larvae display normal initial muscle development, which was followed by rapid loss of muscle integrity and increased expression of the ECM protein laminin 111. Increased laminin 111 appeared to be compensatory because disruption of adhesion to laminin increased muscle degeneration in *spns1^−/−^* larvae. Finally, *spns1^−/−^* larvae exhibit muscle damage caused by sarcolemmal instability that coincides with Tnfsf12-induced MMP-9 expression, but inhibition of Tnfsf12 signaling was not sufficient to maintain muscle integrity in *spns1^−/−^* larvae. These data provide novel insight into the consequences of dysregulated lysosomal pH on muscle development. Although further studies are required to show whether dysregulated lysosomal pH impacts homeostasis in mammals, our data set the stage for these future experiments.

## Data Availability

All data is contained within the article.

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
