# Peer review of "Lysosomal Function Impacts the Skeletal Muscle Extracellular Matrix"

_jdb, 2021, doi:10.3390/jdb9040052_

Round 1

Reviewer 1 Report

Manuscript: Coffey et al. "Lysosomal function impact the skeletal muscle extracellular matrix"

In this manuscript, the authors use zebrafish genetic models and pharmacological inhibition to consider the role of the sphingolipid transporter (spns1) and its downstream mediators in instigating muscle degeneration.  Disruption of spns1 alters expression of ECM proteins, such as laminin 111, leading to morphological abnormalities in muscle at approximately 3.5 dpf due to dysregulation of lysosmal pH.  Additional experiments assess the mechanistic link from Spns1 to Tnfsf12 to excess MMP9. Furthermore, Spns1 deficient fish have an MMP9 membrane integrity problem.

 Data presentation in figures are outstanding in image quality and compelling design.  Analyses are appropriate.  Writing is clear  The paper leads the viewer to conclude that the process is of a complexity that is still at least partially elusive.

-No concerns with study design or parameters.

-No major edits recommended.

Author Response

We thank the reviewer for their time and kind assessment of this manuscript.

Reviewer 2 Report

Dear authors,

Your original research article “Lysosomal Function Impacts the Skeletal Muscle Extracellular Matrix” showed that dysregulated lysosomal pH disrupts muscle homeostasis and causes muscle degeneration. The article informs very well about the main objectives and result of your research. It is well written and easy to read.

I have some recommendation: 

In the discussion section needs a little bit more explanation about the limitations.

  1. None of these findings were validated in human tissues of patients with muscle diseases to further proof your concept of lysosomal dysregulation as a part of the muscle pathology in muscle diseases. Can you please commend this as limitation or recommendation for further studies in the discussion.
  2. Can you explain the translational aspects of your work.
  3. Are there any know disease associated with mutations in Sphingolipid transporter 1 (SPNS1) in humans?

Author Response

Reviewer comments: In the discussion section needs a little bit more explanation about the limitations.

We thank the reviewer for their time and effort in reviewing this manuscript. 

  1. None of these findings were validated in human tissues of patients with muscle diseases to further proof your concept of lysosomal dysregulation as a part of the muscle pathology in muscle diseases. Can you please commend this as limitation or recommendation for further studies in the discussion.
  2. Can you explain the translational aspects of your work.

RESPONSE: We added that although we did not investigate the impact of dysregulated lysosomal pH in human tissues, our experiments in the zebrafish model to set the stage for these future experiments.

3. Are there any know disease associated with mutations in Sphingolipid transporter 1 (SPNS1) in humans?

RESPONSE: There are not actually any published known cases of spns1 in human disease and we have added this to the discussion.

Reviewer 3 Report

The results include a large number of variables in each of the 9 subsections. My comments is to facilitate the understanding and follow-up of the study, it would be appropriate to make a table showing the variables with the over- or under-expression and the statistical significance. 

Author Response

We thank the reviewer for this suggestion and have now included such a table. 

Round 2

Reviewer 2 Report

Dear Authors, congrats to your great work, after including several figures the story sounds now very well.